# Research Progress and Prospects of Molecular Breeding in Bermudagrass (*Cynodon dactylon*)

**DOI:** 10.3390/ijms252413254

**Published:** 2024-12-10

**Authors:** Xiaoyang Sun, Qiang Fu, Yuxiao Song, Xinjie Deng, Yinruizhi Li, Ke Wu, Shuning Li, Jinmin Fu

**Affiliations:** College of Grassland Science, Qingdao Agricultural University, Qingdao 266109, China; sunxiaoyang@qau.edu.cn (X.S.); fqiang010000@163.com (Q.F.); s18606378002@163.com (Y.S.); 17685458206@163.com (X.D.); 202301007@qau.edu.cn (Y.L.); 18214491468@163.com (K.W.); 17669790168@163.com (S.L.)

**Keywords:** bermudagrass, genetic manipulation, stress tolerance, CRISPR/Cas, multi-omics

## Abstract

Bermudagrass (*Cynodon dactylon* L.) is a warm-season grass species of significant ecological and economic importance. It is widely utilized in turf management and forage production due to its resilience to drought, salt, and other environmental stresses. Recent advancements in molecular breeding, particularly through genomics technology and gene editing, have enabled the efficient identification of key genes associated with stress tolerance and turf quality. The use of techniques such as overexpression and CRISPR/Cas has enhanced resistance to drought, salt, cold, and heat, while the application of molecular markers has accelerated the development of superior varieties. The integration of multi-omics, such as genomics, transcriptomics, and proteomics, provides deeper insights into the molecular mechanisms of bermudagrass, thereby improving breeding efficiency and precision. Additionally, artificial intelligence is emerging as a powerful tool for analyzing genomic data, predicting optimal trait combinations, and accelerating breeding processes. These technologies, when combined with traditional breeding methods, hold great potential for optimizing bermudagrass varieties for both turf and forage use. Future research will focus on further integrating these tools to address the challenges of breeding posed by climate change to breeding climate-resilient turf and forage crops.

## 1. Introduction

Bermudagrass (*Cynodon dactylon* L.) is a warm-season grass species with significant ecological and economic value. Due to its excellent drought tolerance, salt tolerance, and disease resistance, it is widely used in turf management and forage production worldwide [1]. It is distributed across many regions in China and globally, particularly excelling in low-maintenance turf construction and soil erosion prevention [2]. Bermudagrass’s strong adaptability and rapid growth make it an ideal forage option for maintaining high yields in harsh environmental conditions [3]. It holds a key position not only in turf management but also in forage production, where its potential is widely recognized, especially in the livestock industry.

The economic value of bermudagrass lies primarily in its diverse uses. As a turf grass, it is commonly applied in parks, sports fields, and gardens, favored for its low maintenance requirements and resilience to stress. As a forage grass, it can maintain stable yields even in harsh environments such as drought and saline–alkaline soils, making it a crucial grass species for livestock development in many regions [4]. As ecological grass, bermudagrass is the most abundant and dominant species in heavy metal-contaminated soils in the southern area of China [5]. Previous studies have demonstrated that bermudagrass has great potential in the remediation of Cd-, Pb-, and polycyclic aromatic hydrocarbon-contaminated soil [6]. Additionally, the genetic diversity of bermudagrass offers great potential for cultivar improvement, enabling the development of superior varieties with greater tolerance to environmental stresses, thus further expanding its range of applications.

With worsening environmental conditions, especially the challenges posed by global climate change, breeding turf and forage varieties with enhanced stress resistance has become a priority in breeding research. Due to its complex genetic background and strong adaptability, bermudagrass exhibits significant genetic and morphological diversity, providing valuable genetic resources for cultivar improvement [7]. Therefore, molecular breeding research on bermudagrass plays a crucial role in enhancing breeding efficiency and improving stress tolerance traits.

This review aims to summarize the latest progress in the molecular breeding of bermudagrass, discussing the improvement of its genetic resources and stress resistance traits, as well as strategies for enhancing turf and forage quality. It highlights the application of gene editing technologies and molecular marker-assisted breeding, analyzes recent key research findings, and provides insights into future breeding directions and challenges, offering scientific evidence and practical guidance.

## 2. Genetic Resources and Genomics of Bermudagrass

### 2.1. Genetic Diversity Research

In recent years, significant progress has been made in the study of the genetic diversity and population structure of bermudagrass germplasm resources based on molecular marker technologies. Using molecular markers such as inter-primer binding site (iPBS) markers and simple sequence repeat (SSR) markers, researchers have assessed bermudagrass genotypes from diploid to hexaploidy, revealing a high level of genetic diversity [8]. This diversity provides valuable resources for the genetic improvement of bermudagrass varieties. Additionally, multivariate analysis under drought stress has further revealed the differences in bermudagrass germplasm resources under different environmental conditions, highlighting the potential application of molecular marker technologies in selecting stress-resistant traits [9].

### 2.2. Whole-Genome Sequencing

Whole-genome sequencing provides critical data for understanding the complex genetic structure of bermudagrass. The bermudagrass genome exhibits a complex polyploid structure, particularly in diploid and tetraploid forms, displaying significant genetic variation [10]. Whole-genome sequencing has revealed remnants of polyploidy, indicating that this species may have undergone diploidization after whole-genome duplication [11]. These genomic data not only offer new insights into the evolutionary history of bermudagrass but also lay a foundation for understanding its relationship with other related grass species.

Furthermore, bermudagrass genome analysis helps breeders improve their turfgrass and forage traits. For example, the ‘Tifton 85’ forage cultivar, improved through traditional breeding techniques, achieved higher digestibility by reducing phenolic acid content [12]. Additionally, bermudagrass genome sequencing revealed its polyploid genome at the chromosomal level, consisting of approximately 6.04 billion base pairs and 30,000 genes. The homology of these genes provides a deeper understanding of their gene expression regulation [11].

### 2.3. Application of Genomic Data

The application of genomic data provides strong support for the molecular breeding of bermudagrass. In the genetic improvement of turfgrass and forage, genomic data significantly enhance breeding efficiency through the development of molecular markers. For example, SSR markers have been widely used to assess bermudagrass genetic diversity and identify markers related to turf quality and stress resistance [13]. These markers show high polymorphism at different ploidy levels, providing a basis for the precise selection of superior varieties with salt and drought tolerance.

Through genome-wide association studies (GWAS), researchers have also identified several genes associated with salt tolerance, such as ethylene-responsive transcription factor (*RAP2-2*) and the cyclic nucleotide-gated ion channel 1 (*CNG1*), which significantly influence bermudagrass adaptation to saline–alkaline environments [11]. The application of these molecular markers has accelerated the breeding of salt-tolerant varieties while enhancing bermudagrass’s adaptability to complex environments.

In summary, we summarized the timeline of research on bermudagrass genetic resources and genomics (Figure 1). The genomic data will play a critical role in the molecular breeding of bermudagrass, especially in the rapid breeding of new bermudagrass varieties with greater adaptability and higher quality. Genomic technologies provide powerful tools for future breeding efforts.

## 3. Molecular Basis of Stress Resistance in Bermudagrass

### 3.1. Identification and Expression Analysis of Drought Resistance Genes

The regulation of drought tolerance in bermudagrass involves several key genes of which the expression patterns vary across different organs. Notably, DEGs encoding starch synthesis-related enzymes all showed the highest expression level in the rhizome and were closely related to drought tolerance under drought conditions [14]. Transcriptome analysis has revealed multiple pathways related to plant hormone signal transduction and gene regulation, which help bermudagrass maintain normal physiological functions under drought stress [14].

Further research through mitochondrial genome sequencing of bermudagrass hybrids with *C. transvaalensis* has identified potential drought-related genes. These genes are closely associated with responses to drought stress [15]. In *C. dactylon*, *CdDHN4*, antioxidant genes *Cu/ZnSOD* and *APX* lead to higher antioxidant activities to improve its drought tolerance [16]. The findings provide new molecular insights into the exploration of drought tolerance genes in bermudagrass and lay a foundation for molecular breeding strategies (Figure 2).

### 3.2. Identification and Expression Analysis of Salt Tolerance Genes

Bermudagrass can tolerate 0.6–1.0% salt content; salinity stress induced the selective transport of K^+^ over Na^+^ from roots to leaves and selectively secreting Na^+^ via the leaf salt gland [17]. Biochemical changes include a change in the osmotic substances and antioxidative enzymes, like soluble protein content and proline levels, superoxide dismutase (SOD), and peroxide dismutase (POD) activities increase [18]. However, salt stress sensitivity increased in *CdWRKY2* overexpression lines of bermudagrass, which showed growth inhibition of the root system [19].

Salt tolerance in bermudagrass has been revealed primarily through genome-wide association studies (GWAS) and gene expression analyses, like *RAP2-2*, *CNG channel*, and the leucine-rich repeat receptor-like protein kinase gene *F14D7.1* [11]. Meanwhile, 536 miRNAs were found to be involved in the salt stress response, and *miRNA171f* could increase the salt tolerance by improving the photosynthetic performance of bermudagrass [20,21]. Studies have shown that several genes related to ion balance and water regulation are significantly upregulated under salt stress, aiding bermudagrass in maintaining normal growth in high-salt environments [19]. For instance, high-density genetic maps have revealed chromosome rearrangements and the genome evolution process in bermudagrass, providing a basis for further investigation into salt tolerance traits [22,23].

Additionally, molecular marker technology has improved the efficiency of screening for salt tolerance traits. Researchers have identified several salt tolerance-related genes using SSR markers and have accelerated the breeding of salt-tolerant varieties through marker-assisted selection. This strategy, which combines molecular markers and genetic engineering, not only enhances bermudagrass’s adaptability to saline–alkaline environments but also provides potential genetic resources for future improvements in turfgrass and forage (Figure 2).

### 3.3. Identification and Expression Analysis of Cold Tolerance Genes

Cold tolerance in bermudagrass has been primarily revealed through transcriptome and gene regulatory pathway analyses. Under cold stress, genes such as *HSP70/90* and *HsfA3/A8* were significantly upregulated [24]. These genes regulate antioxidant systems and chlorophyll synthesis, helping bermudagrass maintain growth under low-temperature conditions [25]. Additionally, the *CdtCIPK21* gene from the *CIPK* family shows enhanced expression under low temperatures, regulating cold tolerance genes like *CdDREB1A* and *CdLEA3*, significantly improving cold tolerance [26].

These genes can be used for the selection of cold tolerance traits and provide new directions for molecular breeding. The application of such molecular marker technologies has accelerated the development of cold-tolerant bermudagrass varieties.

### 3.4. Identification and Expression Analysis of Heat Tolerance Genes

Although bermudagrass is a warm-season grass, heat stress is a limiting factor for its growth and development. Previously, studies have found that some genes can improve the heat stress tolerance of bermudagrass. In *C. transvaalensis*, the genome-wide analysis of the *HSP* genes family and expression patterns of *HSP20* gene in response to heat stress [27]. In addition, *CtHsfA2b* was also identified from African bermudagrass exhibiting a rapid response to high temperature in Arabidopsis [28]. Meanwhile, RNA sequencing analysis found that the *CdF-box* gene in bermudagrass (the E3 ubiquitin ligase-related gene) played an important role in improving the heat stress tolerance of transgenic Arabidopsis. Notably, *CdmiRNA159a*, *CdmiRNA160a*, and *CdmiRNA164f* and their target genes (*CdGAMYB*, *CdARF17*, and *CdNAC1*, respectively) play a critical role in response to heat stress in bermudagrass [29].

## 4. Molecular Breeding of Bermudagrass Traits

### 4.1. Genetic Improvement of Turf Traits in Bermudagrass

Key goals in bermudagrass breeding include improving turf traits such as grass density, color, and wear resistance. Through GWAS and weighted gene co-expression network analysis (WGCNA), researchers have identified the β-glucosidase 31 (*CdBGLU31*) gene, which is closely related to plant height and auxin content in bermudagrass. This finding provides a clear genetic basis for improving turf appearance traits [30]. Then, a high-density genetic map was constructed and identified *CtQCTER2* gene loci associated with turf establishment speed, providing effective molecular tools for improving grass density and establishment rate [31].

Additionally, transcriptome analysis revealed that changes in chlorophyll synthesis and antioxidant capacity under different temperature conditions directly impact turf color in bermudagrass [25]. Through ISSR marker analysis, studies have revealed the genetic diversity of bermudagrass germplasm, providing a solid foundation for selecting and screening varieties with stress-resistant traits, such as wear resistance [32].

Xu et al. summarized the application of gene transformation technology in enhancing bermudagrass turf durability, particularly in improving salt resistance and tolerance to environmental stress [33]. These studies have laid the molecular breeding foundation for improving the appearance and durability of bermudagrass turf, promoting its widespread use in high-quality turf management (Figure 2).

### 4.2. Optimization of Bermudagrass Growth Cycle

The optimization of bermudagrass’s growth cycle has significantly improved growth rate, mowing tolerance, and disease resistance through molecular marker technology. Researchers, through the construction of high-density genome-wide association maps, have identified several gene loci associated with rapid growth and the turf establishment rate. These genes provide reliable molecular tools for improving bermudagrass growth rate [34]. Additionally, gene transformation technology, by introducing the *CdHEMA1* gene, has significantly enhanced the photosynthetic efficiency and mowing tolerance of bermudagrass [33].

In terms of disease resistance, polyploid studies have shown that polyploid bermudagrass exhibits stronger disease resistance. Gulsen et al. discovered that bermudagrass plants with different ploidy levels show significant differences in disease resistance, providing valuable molecular markers for disease-resistant breeding [34]. Genome studies of the Cynodon species have also identified key genes related to disease resistance, particularly the *HSP* genes family involved in responses to salt stress and pathogen infections. These findings provide important resources for breeders to improve disease-resistance traits.

### 4.3. Molecular Improvement of Stress Resistance in Bermudagrass

Drought tolerance, salt tolerance, and cold tolerance are key focuses in bermudagrass breeding efforts. Through genetic diversity research, researchers have used RAPD markers to analyze bermudagrass and other salt-tolerant plants, identifying potential molecular markers associated with salt tolerance [25,35]. Additionally, the enhanced expression of the *CdtCIPK21* gene under cold stress significantly improves cold tolerance in bermudagrass, while its negative regulatory effect under salt stress provides valuable genetic resources for the improvement of both salt and cold tolerance traits [25]. In addition, *CdWRKY2* plays an important role in regulating cold stress response via activating the sucrose-related gene *CdSPS1* and the cold response-related gene *CdCBF1* in bermudagrass [26].

In terms of drought tolerance, SSR marker technology has helped researchers identify molecular markers related to cadmium tolerance and drought tolerance, providing important references for the breeding of drought-tolerant varieties [2,36]. Yu et al. identified multiple QTL loci associated with winter survival and leaf fire tolerance through QTL mapping, suggesting that molecular marker technology can be used to simultaneously improve both cold and drought tolerance in bermudagrass [37].

Fang et al. revealed chromosome rearrangements and genome evolution in bermudagrass through genome sequencing, providing valuable genomic information for the study and improvement of stress-resistant traits [22]. Yu et al. further explored the genetic basis of cold and drought tolerance, proposing that by integrating molecular marker technologies, multiple stress-resistant traits can be improved simultaneously, offering effective strategies for future breeding work [37].

### 4.4. Integration of Turf and Forage Bermudagrass Breeding

Although the breeding goals for turf and forage bermudagrass differ, there are many commonalities in molecular improvement. Turf bermudagrass primarily focuses on high density, good color, and wear resistance, while forage bermudagrass emphasizes increased biomass and nutrient content, such as crude protein and cellulose. Using SSR and ISSR markers, research has revealed the rich genetic diversity of bermudagrass, providing a foundation for the simultaneous improvement of bermudagrass traits [38,39].

Molecular markers not only help improve the stress resistance of turf bermudagrass (such as salt and drought tolerance) but also support the selection of forage bermudagrass varieties. For example, Xiang et al. identified gene loci related to turf quality through salt stress experiments, which are also helpful in forage improvement [10]. Tiwari et al. pointed out that bermudagrass’s core germplasm collection contains multiple stress-resistant traits, which can be efficiently selected through molecular markers [40].

Although the breeding directions for turf and forage bermudagrass differ, their molecular strategies for dealing with environmental stress and improving genetic diversity share similarities (Figure 2). In addition, microorganism plant growth-promoting rhizobacteria (PGPR) and arbuscular micorrhizal fungi (AMF) can improve bermudagrass stress resistance and quality. Applications of PGPR with a half rate of fertilizer yielded similar bermudagrass forage biomass and quality results as that of a full rate of nitrogen [41]. Previously studied arbuscular AMF can help bermudagrass grow in a lead–zinc mine wasteland [42]. The technologies can improve turf quality while optimizing forage yield, promoting bermudagrass’s use in multiple fields.

## 5. Application of Gene Editing Technology in Bermudagrass Breeding

### 5.1. Application of CRISPR/Cas Technology in Stress Resistance Improvement

CRISPR/Cas gene editing technology provides a powerful tool for improving the stress resistance of bermudagrass. By precisely modifying key genes, researchers can significantly enhance bermudagrass’s tolerance to drought, salt, and cold. Guo et al. demonstrated that by editing genes related to salt and drought tolerance, bermudagrass’s survival ability in extreme environments, particularly in saline–alkaline and drought conditions, was greatly enhanced [21]. This discovery opens new avenues for the application of bermudagrass in harsh climates.

Furthermore, Fang et al. showed that gene editing technology has great potential in enhancing drought tolerance in bermudagrass. By editing genes closely related to drought tolerance, bermudagrass’s ability to withstand drought stress was significantly improved [22]. Similarly, Liu et al. used CRISPR technology to enhance bermudagrass’s cold tolerance, significantly increasing its survival in low-temperature environments [25].

The application of CRISPR technology has greatly advanced the improvement of stress-resistant traits in bermudagrass, allowing it to adapt more broadly to harsh environments worldwide.

### 5.2. Application of CRISPR/Cas Technology in Turf Quality and Forage Improvement

CRISPR/Cas technology also plays a significant role in improving turf quality. By editing genes related to grass density, height, and growth patterns, researchers were able to significantly enhance the appearance of bermudagrass turf, resulting in higher grass density, more uniform coverage, and better mowing tolerance [38]. Xu et al. further optimized the *CdHEMA1* transformation system to improve bermudagrass’s photosynthetic efficiency and mowing tolerance, laying the foundation for the long-term health of bermudagrass turf [33].

In addition to improving turf quality, CRISPR and overexpression techniques have also achieved remarkable progress in bermudagrass. By editing genes related to crude protein and cellulose content, we will significantly improve the nutritional value of bermudagrass, making it a more efficient forage option [43]. This improvement not only increased the biomass of bermudagrass but also enhanced its adaptability to extreme environments, achieving the dual goals of improving both turf and forage traits.

Jewell et al.’s research further supports this, pointing out that by combining molecular markers and gene editing, bermudagrass can more efficiently achieve the dual improvement of turf and forage quality, thereby promoting its broad application in various fields [39].

## 6. Marker-Assisted Breeding in Bermudagrass

### 6.1. QTL Mapping and Applications in Stress-Resistant Trait Selection

Quantitative trait locus (QTL) mapping has important applications in the selection of stress-resistant traits in bermudagrass. Through QTL analysis, researchers have identified gene loci associated with drought, salt, cold, and heat tolerance, providing key molecular markers for molecular breeding [44]. Fang et al. revealed multiple gene loci related to drought, salt, and flood tolerance in bermudagrass through transcriptomic and proteomic analysis, laying the foundation for subsequent trait improvement [22]. Hu et al. further identified several QTLs related to cold tolerance in bermudagrass, providing molecular evidence for cold-tolerant breeding [45].

Guo et al. identified several drought-tolerant genes through QTL mapping that significantly influence plant growth under drought conditions, providing target genes for drought-tolerant breeding [21]. Tiwari et al. used SSR marker technology to analyze the core germplasm collection of bermudagrass and identified multiple QTLs associated with stress resistance, laying the genetic foundation for further stress-resistant breeding [40].

Additionally, Xiang et al. identified loci related to salt tolerance through QTL analysis, which can be used to develop salt-tolerant varieties, thereby improving bermudagrass’s adaptability in saline–alkaline soils [10]. The identification and application of these QTLs provide crucial technical support and strategies for molecular breeding efforts.

### 6.2. Development and Application of Molecular Markers in Breeding

The development and application of molecular markers in bermudagrass breeding are based on genomic data, enabling the precise selection of gene loci associated with desirable traits, thus enhancing breeding efficiency. Wang et al. developed a set of molecular markers based on genomic data, which were applied to improve salt tolerance in bermudagrass. Through genome-wide association studies (GWAS), they identified several salt tolerance-related gene loci, such as *RAP2-2*, *CNG channel*, and the probable leucine-rich repeat receptor-like protein kinase (*F14D7.1*) [11]. These markers provide precise molecular tools for further salt-tolerant breeding.

Serba et al. used 51 pairs of SSR markers to conduct DNA analysis on 21 bermudagrass breeding materials and identified multiple polymorphic markers related to drought tolerance. These markers significantly enhanced the ability to distinguish different drought-tolerant varieties, providing important references for advancing stress-resistant breeding [46]. Huang et al. revealed the phenomenon of chloroplast genome rearrangement in bermudagrass, and through genomic data analysis, they discovered a series of potential gene markers that contribute to improving bermudagrass adaptability and stress resistance [47].

Additionally, Grossman et al. studied the germplasm resources of bermudagrass at different ploidy levels, using flow cytometry to measure chromosome numbers and SSR markers to assess genetic diversity. Their research found that ploidy levels were significantly associated with bermudagrass yield and nutritional quality, and the marker data contributed to optimizing forage breeding strategies [48].

Finally, Wang et al. analyzed the heat shock transcription factor A2 (*CtHsfA2*) gene in African bermudagrass and found that this gene improves heat tolerance through transcriptional regulation under heat stress, providing new molecular marker resources for developing heat-tolerant varieties [24]. The development and application of these molecular markers have greatly improved breeding efficiency for stress resistance and turf quality in bermudagrass.

### 6.3. Application of Epigenetic Regulation in Breeding

The application of epigenetics in bermudagrass breeding has shown significant value, particularly in trait improvement. Epigenetic regulation, such as DNA methylation and histone modification, can regulate gene expression without altering the DNA sequence, thereby influencing a plant’s ability to adapt to environmental stresses. Hu et al. studied the role of ethylene in cold stress and found that ethylene enhanced cold tolerance in bermudagrass by regulating the antioxidant system and photosynthesis [45]. This epigenetic regulation offers new insights for improving cold tolerance traits.

Xie et al. investigated the epigenetic changes in bermudagrass under cadmium (Cd) heavy metal stress and demonstrated that histone methylation modifications can regulate the expression of resistance genes, thereby enhancing bermudagrass’s cadmium tolerance [49]. This discovery provides new molecular tools for breeding against heavy metal pollution.

In addition, Jewell et al. highlighted that epigenetic regulation plays an important role in maintaining and improving the genetic diversity of bermudagrass, especially when dealing with various biotic and abiotic stresses. This regulatory mechanism helps improve bermudagrass adaptability [39].

Taghizadeh et al. further pointed out that epigenetic regulation mechanisms are equally important in salt tolerance breeding, helping plants maintain normal physiological activities under salt stress and enhancing their adaptability [50]. Pudzianowska and Baird’s research also showed that epigenetics has the potential to improve bermudagrass turf traits, such as density and disease resistance, offering a new approach to enhancing turf quality [15].

## 7. Future Outlook

### 7.1. Collaborative Application of Molecular Breeding and Traditional Breeding

The breeding strategies for bermudagrass are gradually combining traditional breeding with molecular breeding to improve breeding efficiency and achieve precise improvement of important traits. Traditional breeding methods rely on the selection of morphological and physiological characteristics, while molecular breeding enables trait selection based on genomic information and molecular markers [51]. The combination of these two approaches can significantly accelerate the breeding process. By molecular marker technologies such as ISSR, researchers can identify genetic markers associated with traits at the early stages of breeding, providing breeders with more precise tools for selection [35].

Previously study revealed the complete genome assembly of bermudagrass makes genome-based molecular marker development possible [52]. This supports germplasm selection and marker-assisted selection in traditional breeding, helping to achieve the faster development of superior varieties.

Furthermore, Fang et al.’s research showed that transcriptome analysis can identify genes related to stress-resistant traits, and these genes can be further utilized in molecular breeding through traditional crossing and selection methods to enhance stress resistance [52,53]. The combination of traditional breeding strategies, such as hybrid breeding, with marker-assisted selection can improve both the stress resistance and turf quality of bermudagrass while increasing breeding efficiency [54].

The integration of these two strategies not only accelerates the selection of target traits through molecular technology but also enriches the germplasm pool through traditional breeding. This ultimately leads to comprehensive improvement of the target traits.

### 7.2. The Potential of Multi-Omics and AI in Bermudagrass Breeding

Emerging technologies, such as multi-omics and artificial intelligence (AI), are increasingly recognized for their potential in bermudagrass breeding. By integrating data from genomics, transcriptomics, proteomics, and other omics fields, breeders can gain a comprehensive understanding of the molecular mechanisms in bermudagrass, thereby improving breeding efficiency. Liu et al. used RNA sequencing to analyze gene expression in bermudagrass under different stress conditions, revealing genome-level molecular mechanisms for coping with environmental stress, which provides a strong foundation for the application of multi-omics in bermudagrass breeding [28].

AI technology, including machine learning (ML) and deep learning (DL), has significantly advanced the development of superior bermudagrass varieties. Using stereovision and ML algorithms, researchers have effectively estimated the aboveground biomass and vegetation coverage of forage Bermudagrass [55]. In addition, DL networks were used to detect and estimate the coverage rate of weed in bermudagrass turf [56]. Moreover, with the development of AI technology, the use of AI for big data analysis and model prediction can accelerate the processing of genomic data. Zhang et al. analyzed proteomic data from the stems and stolons of bermudagrass, providing data support for the molecular improvement of turf traits [57]. Shi et al. demonstrated that the combination of AI and multi-omics can help breeders predict and select the best trait combinations, improving bermudagrass’s stress resistance and turf quality [58].

These studies indicate that integrating multi-omics and artificial intelligence technologies will greatly enhance the efficiency and precision of bermudagrass breeding, offering broad application prospects for future breeding strategies (Figure 3).

## 8. Conclusions

Recently, significant progress has been made in the molecular breeding of bermudagrass, particularly in the improvement of stress resistance traits and turf quality. In the future, combining genomics, molecular markers, and gene editing technologies will further deepen the understanding of the molecular mechanisms in bermudagrass. The introduction of AI technology is expected to optimize breeding decisions through big data analysis, accelerating the process of variety improvement. The fusion of these emerging technologies will drive bermudagrass breeding toward greater efficiency and modernization.

## Figures and Tables

**Figure 1 ijms-25-13254-f001:**
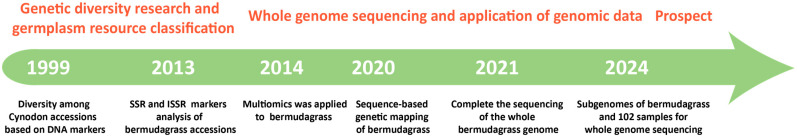
Timeline of research on bermudagrass genetic resources and genomics.

**Figure 2 ijms-25-13254-f002:**
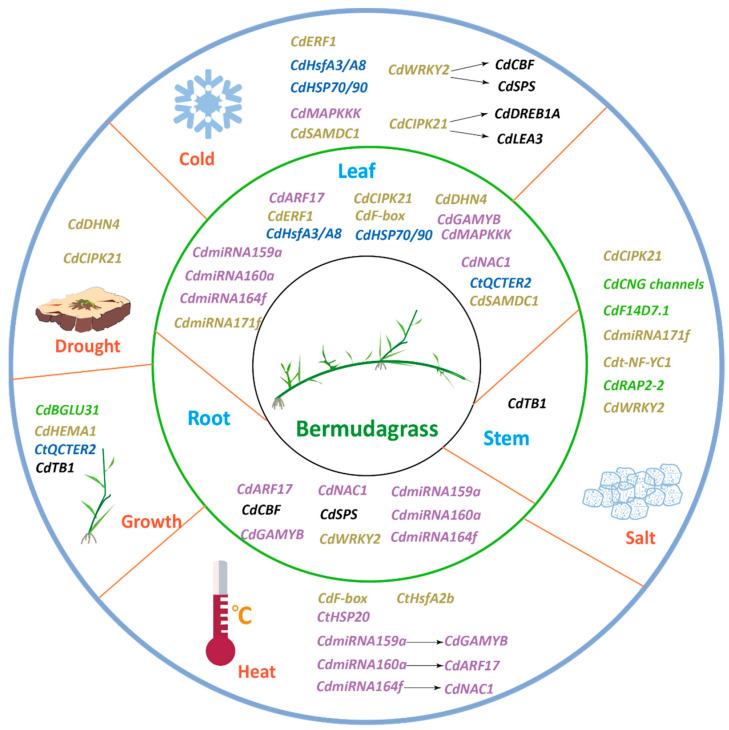
Genes related to environmental abiotic stress and growth in bermudagrass. The sky-blue fonts represent genes that were identified by transcriptome or QTL mapping. The light-green fonts represent genes that were identified by GWAS. The gray-green fonts represent genes that were identified by the genetically verified.

**Figure 3 ijms-25-13254-f003:**
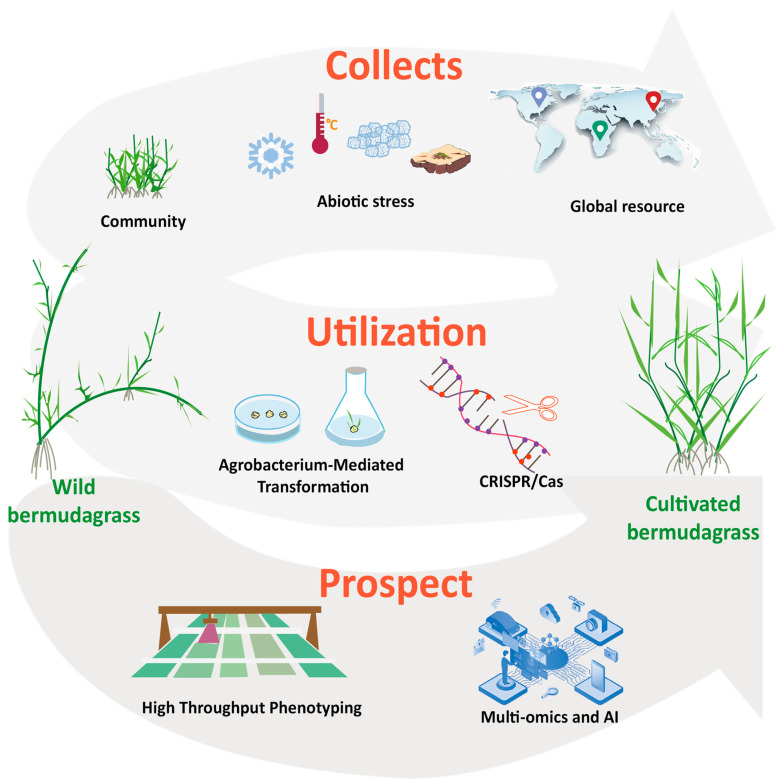
Infographic showing the workflow to optimize the collection and utilization of wild bermudagrass and to promote the genetic improvement of cultivated bermudagrass drawing by Adobe Illustrator.

## Data Availability

Data sharing is not applicable to this article as no datasets were generated or analyzed during the current study.

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
