# Peer review of "Research Progress and Prospects of Molecular Breeding in Bermudagrass (Cynodon dactylon)"

_ijms, 2024, doi:10.3390/ijms252413254_

Round 1

Reviewer 1 Report

Comments and Suggestions for Authors

Bermudagrass (Cynodon dactylon L.) is a warm-season grass species with significant ecological and economic value. This review aims to summarize the latest progress in the molecular breeding of bermudagrass, highlight the application of gene editing technologies and provides insights into future breeding directions and challenges. However, there are some issues should be solved.

1. Efforts should be made to revise the manuscript as a deep comprehensive analyzed review rath than just a collection of published data.

2. There are lots of citation errors "Error! Reference source not found." in the whole context, for instance Line30, 32, 34, etc., the authors should double check the whole manuscript and add the citation of the references.

3. In Fig1, the top boldface letters seemed to be compressed. Were they?

4. In line 98, “…and the cyclic nucleotide-gated ion channel 1(CNG channels)” should be “…and the cyclic nucleotide-gated ion channel 1(CNG1)”;

5. Similarly, in line128, “like RAP2-2, CNG channel, and the probable leucine-rich repeat receptor-like protein kinase (F14D7.1) genes” should be “like RAP2-2, CNG1, and the leucine-rich repeat receptor-like protein kinase gene F14D7.1”.

6. In line122, “…antioxidant genes Cu/ZnSOD and APX”, these genes need to be italicized as Cu/ZnSOD and APX.

7. In line169, “…that CdF-box gene in bermudagrass (the E3 ubiquitin ligase-related gene)” should be “…that CdF-box gene (the E3 ubiquitin ligase-related gene) in bermudagrass…”, and the authors are suggested to give the specific name of the gene.

Comments on the Quality of English Language

Bermudagrass (Cynodon dactylon L.) is a warm-season grass species with significant ecological and economic value. This review aims to summarize the latest progress in the molecular breeding of bermudagrass, highlight the application of gene editing technologies and provides insights into future breeding directions and challenges. However, there are some issues should be solved.

1. Efforts should be made to revise the manuscript as a deep comprehensive analyzed review rath than just a collection of published data.

2. There are lots of citation errors "Error! Reference source not found." in the whole context, for instance Line30, 32, 34, etc., the authors should double check the whole manuscript and add the citation of the references.

3. In Fig1, the top boldface letters seemed to be compressed. Were they?

4. In line 98, “…and the cyclic nucleotide-gated ion channel 1(CNG channels)” should be “…and the cyclic nucleotide-gated ion channel 1(CNG1)”;

5. Similarly, in line128, “like RAP2-2, CNG channel, and the probable leucine-rich repeat receptor-like protein kinase (F14D7.1) genes” should be “like RAP2-2, CNG1, and the leucine-rich repeat receptor-like protein kinase gene F14D7.1”.

6. In line122, “…antioxidant genes Cu/ZnSOD and APX”, these genes need to be italicized as Cu/ZnSOD and APX.

7. In line169, “…that CdF-box gene in bermudagrass (the E3 ubiquitin ligase-related gene)” should be “…that CdF-box gene (the E3 ubiquitin ligase-related gene) in bermudagrass…”, and the authors are suggested to give the specific name of the gene.

Author Response

Responses to editor and reviewers

Many thanks for your reviews. All comments from the editor and this reviewer have been considered, and we have addressed all of them in the revision. Our responses to individual comments are listed below. The revised lines were marked in red font in the manuscript.

Responses to reviewer #1:

1. Efforts should be made to revise the manuscript as a deep comprehensive analyzed review rath than just a collection of published data.

Responses:

We revised as suggested in the revised manuscript. We rewrote the abstract and the conclusion. We have made significant adjustments to the manuscript (revised line 10-24, 422-430).

2.There are lots of citation errors "Error! Reference source not found." in the whole context, for instance Line30, 32, 34, etc., the authors should double check the whole manuscript and add the citation of the references.

Responses:

We have made the revisions as suggested in the revised manuscript.

3. In Fig1, the top boldface letters seemed to be compressed. Were they?

Responses:

Yes, the top boldface letters were compressed. We revised Fig1.

4. In line 98, “…and the cyclic nucleotide-gated ion channel 1(CNG channels)” should be “…and the cyclic nucleotide-gated ion channel 1(CNG1)”;

Responses:

We revised as suggested (revised line 100).

5. Similarly, in line128, “like RAP2-2, CNG channel, and the probable leucine-rich repeat receptor-like protein kinase (F14D7.1) genes” should be “like RAP2-2, CNG1, and the leucine-rich repeat receptor-like protein kinase gene F14D7.1”.

Responses:

We revised the manuscript as suggested (revised line 136-137).

6. In line122, “…antioxidant genes Cu/ZnSOD and APX”, these genes need to be italicized as Cu/ZnSOD and APX.

Responses:

We revised the manuscript as suggested (revised line 122-123).

7. In line169, “…that CdF-box gene in bermudagrass (the E3 ubiquitin ligase-related gene)” should be “…that CdF-box gene (the E3 ubiquitin ligase-related gene) in bermudagrass…”, and the authors are suggested to give the specific name of the gene.

Responses:

We revised the manuscript as suggested (revised line 170-174).

Reviewer 2 Report

Comments and Suggestions for Authors

Please resubmit the manuscript.

Author Response

Responses to editor and reviewers

Many thanks for your reviews. All comments from the editor and this reviewer have been considered, and we have addressed all of them in the revision. Our responses to individual comments are listed below. The revised lines were marked in red font in the manuscript.

Responses to reviewer #2:

1. Abstract: This section is well presented.

Responses:

We have rewritten the abstract (revised line 10-24).

2. There are too many keywords in the manuscript title. For example, genetic manipulation can replace molecular breeding.

Responses:

We revised as suggested (revised line 25).

3. In my opinion, some information can be added about the phytoremediation potential of Bermuda grass (Cynodon dactylon (L.) pers.) in soils.

Responses:

As ecological grass, bermudagrass was the most abundant and dominant species at heavy metal-contaminated soils in the south of area, China. Previously studies demonstrated that bermudagrass has great potential in the remediation of Cd, Pb and polycyclic aromatic hydrocarbons contaminated soil (revised line 41-44).

4. Each Figure and its explanations must be entered in a Table according to MDPI rules.

Responses:

We revised the manuscript as suggested.

5. More updated references need to be added.

Responses:

We revised as suggested in the revised manuscript.

6. It is an original graphical workflow. Please clarity the software used to generate it.

Responses:

We revised the manuscript as suggested (revised line 371).

7. Some AI technologies need to be mentioned and explained in more detail.

Responses:

AI technology, including machine learning (ML) and deep learning (DL), has significantly advanced the development of superior bermudagrass varieties. Using stereovision and ML algorithms, researchers have effectively estimated the aboveground biomass and vegetation coverage of forage Bermudagrass. In addition, DL networks were used to detect and estimate coverage rate of weed in bermudagrass turf (revised line 407-410).

8. Please rephrase more concisely! A one-paragraph conclusion is sufficient in my opinion.

We revised the manuscript as suggested (revised line 422-430).

9. Please check the list carefully. All references in the manuscript must be included 
in the reference list. Please take note of the suggestion to add more references.

Responses:

We revised as suggested in the revised manuscript.

Reviewer 3 Report

Comments and Suggestions for Authors

The paper represents a well structured sumarry of what is known on genetic improvemen of bermudagrass.

I have only found some minor points.

- I have missed a general view of what advances in GMO and CRISPR have been performed. Is there any commercial GMO or CRISPR bermuda grass in the market? Which is the modified trait? Where is being planted? Please include this information.

Salt tolerance (line 26): The information on salt tolerance is too concise and misses some essential points. What is known on the mechanism of salt tolerance on Bermuda grass? Which is the level of tolerance? The strategy is accumulating salt in leaves or extruding through the roots? Or increasing potassium uptake? Or increasing the amount of osmolytes? Please, clarify all these points.

- Is there anything known on association of Bermuda grass with Plant growth promoting rhizobacteria or Arbuscular micorrhizal fungi? is this an strategy for improvement? Please comment.

Author Response

Responses to editor and reviewers

Many thanks for your reviews. All comments from the editor and this reviewer have been considered, and we have addressed all of them in the revision. Our responses to individual comments are listed below. The revised lines were marked in red font in the manuscript.

Responses to reviewer #3:

I have only found some minor points.

I have missed a general view of what advances in GMO and CRISPR have been performed. Is there any commercial GMO or CRISPR bermudagrass in the market? Which is the modified trait? Where is being planted? Please include this information.

Responses:

So far, there is no CRISPR and overexpression bermudagrass in the market. By editing genes related to grass density, height, and growth patterns, researchers were able to significantly enhance the appearance of bermudagrass turf, resulting in higher grass density, more uniform coverage, and better mowing tolerance. In addition to improving turf quality, CRISPR and overexpression techniques have also achieved remarkable progress of bermudagrass (revised line 10-24, 289-290).

2. Salt tolerance (line 26): The information on salt tolerance is too concise and misses some essential points. What is known on the mechanism of salt tolerance on Bermuda grass? Which is the level of tolerance? The strategy is accumulating salt in leaves or extruding through the roots? Or increasing potassium uptake? Or increasing the amount of osmolytes? Please, clarify all these points.

Responses:

Bermudagrass can tolerate 0.6-1.0% salt content, salinity stress induced the selective transport of K+ over Na+ from roots to leaves and selectively secreting Na+ via leaf salt gland. Biochemical changes include a change in the osmotic substances and antioxidative enzymes, like soluble protein content and proline levels, superoxide dismutase (SOD) and peroxide dismutase (POD) activities increase. However, salt stress sensitivity increased in CdWRKY2 overexpression lines of bermudagrass, which showed growth inhibition of the root system (revised line 128-134).

3. Is there anything known on association of Bermuda grass with Plant growth promoting rhizobacteria or Arbuscular micorrhizal fungi? is this an strategy for improvement? Please comment.

Responses:

Microorganism plant growth-promoting rhizobacteria (PGPR) and arbuscular micorrhizal fungi (AMF) can improve bermudagrass stress resistance and quality. Applications of plant growth-promoting rhizobacteria (PGPR) with a half rate of fertilizer yielded similar bermudagrass forage biomass and quality results as that of a full rate of nitrogen. Previously study arbuscular micorrhizal fungi (AMF) can help bermudagrass growing in a lead–zinc mine wasteland (revised line 252-256).

Round 2

Reviewer 2 Report

Comments and Suggestions for Authors

Well done! Accept in present form.